# The Protective Efficacy of Single-Dose Nasal Immunization with Cold-Adapted Live-Attenuated MERS-CoV Vaccine against Lethal MERS-CoV Infections in Mice

**DOI:** 10.3390/vaccines11081353

**Published:** 2023-08-10

**Authors:** Heejeong Seo, Yunyueng Jang, Dongmi Kwak

**Affiliations:** 1PioneerVaccine, Inc., Chungnam National University, Daejeon 34134, Republic of Korea; hjseo2727@gmail.com; 2College of Veterinary Medicine, Kyunpook National University, Daegu 41566, Republic of Korea

**Keywords:** MERS-CoV, live-attenuated, vaccine, cold-adapted

## Abstract

Middle East respiratory syndrome coronavirus (MERS-CoV) causes severe diseases in humans. Camels act as intermediate hosts for MERS-CoV. Currently, no licensed vaccine is available for this virus. We have developed a potential candidate vaccine for MERS-CoV using the cold adaptation method. We cultivated the vaccine in Vero cells at temperatures as low as 22 °C. This live-attenuated vaccine virus showed high attenuation levels in transgenic mice with the MERS-CoV human receptor, dipeptidyl peptidase 4 (DPP4) (K18-hDPP4). The inoculated K18-hDPP4 mice exhibited no clinical signs such as death or body weight loss. Furthermore, no traces of infectious virus were observed when the tissues (nasal turbinate, brain, lung, and kidney) of the K18-hDPP4 mice infected with the cold-adapted vaccine strain were tested. A single intranasal dose of the vaccine administered to the noses of the K18-hDPP4 mice provided complete protection. We did not observe any deaths, body weight loss, or viral detection in the tissues (nasal turbinate, brain, lung, and kidney). Based on these promising results, the developed cold-adapted, attenuated MERS-CoV vaccine strain could be one of the candidates for human and animal vaccines.

## 1. Introduction

Middle East respiratory syndrome coronavirus (MERS-CoV) was first reported in Saudi Arabia in 2012 [1]. Patients infected with MERS-CoV experience symptoms such as fever, cough, shortness of breath, pneumonia, and diarrhea, with an approximate mortality rate of 35% [2,3,4]. It is believed that MERS-CoV originated from bats [5,6] and is most likely transmitted to humans through dromedary camels in the Middle East, Africa, and south Asia [7,8,9]. Human-to-human transmission of MERS-CoV is rare but can occur among close contacts in a hospital setting [10]. The genome of MERS-CoV consists of a single-stranded positive-sense RNA and encodes four structural proteins: spike (S), envelope (E), membrane (M), and nucleocapsid (N). It also contains open reading frame (ORF) 1a and the ORF 1b genes that encode 16 non-structural proteins (nsp 1–16), as well as the ORF 3, 4a, 4b, 5, and 8b genes that encode 5 accessory proteins [11,12,13]. Among these proteins, the S protein is located on the surface of MERS-CoV and plays a key role in initiating infection by binding to its receptor, human dipeptidyl peptidase 4 (hDPP4), which is present on susceptible cells [14,15]. Efforts have been made to develop a vaccine for MERS-CoV, but no approved vaccine for human use is currently available. Vaccines currently in clinical trials (Phase 1) include a DNA vaccine that expresses the S gene of MERS-CoV [16], an adenovirus-based ChAdOx1 vaccine that expresses the S gene of MERS-CoV [17], and a vaccine based on a viral vector that expresses the S gene of MERS-CoV [18]. 

In this study, we developed a live-attenuated MERS-CoV vaccine strain using the cold adaptation method in Vero cells at temperatures as low as 22 °C. This vaccine strain demonstrated high attenuation levels in K18-hDPP4 mice, which express the human form of the DPP4 receptor. The vaccine’s efficacy was evaluated in these K18-hDPP4 mice.

## 2. Materials and Methods

### 2.1. The Mouse and Virus

The human dipeptidyl peptidase 4 (hDPP4)-transgenic mice (K18-hDPP4) (5-week-old) used in this study were obtained from Dr. Paul B. McCray Jr. from the University of Iowa, Iowa City, IA, USA [19]. The mice were fed with feed and water. The MERS-CoV strain used in the experiments (EMC2012) was obtained from Drs. Bart Haagmans and Ron Fouchier from the Erasmus Medical Center, The Netherlands. The virus was cultured in Vero-E6 cells from the American Type Culture Collection (Manassas, VA, USA) within a biosafety level 3 (BSL3) laboratory facility approved by the Korean government. Additionally, a Korean isolate from 2015 (Korean MERS-CoV/2015) was provided by the Korean Centers for Disease Control and Prevention (KCDC). The experiments involving MERS-CoV were conducted inside of a biosafety level 3 (BSL3) facility.

### 2.2. Generation of Cold-Adapted Live-Attenuated MERS-CoV Vaccine Virus

The cold adaptation of MERS-CoV (EMC2012) was conducted following the described procedure [20]. The virus was adapted from 37 °C to 22 °C using Vero cells cultured in MEM, containing bovine serum albumin (BSA) (1.5%) obtained from Rocky Mountain Biologicals (Missoula, MT, USA) and 1× antibiotic-antimycotic solution (Sigma, St. Louis, MO, USA). The infected Vero cells were monitored for cytopathic effects (CPE), and once CPEs were observed, the virus was further adapted to the next lower temperature. It typically took approximately 3–10 days for CPEs to become apparent in the infected cells. Once MERS-CoV successfully replicated at 22 °C for more than five passages (>passage 5), it was selected for use in the vaccine experiment. The cold-adapted MERS-CoV strain cultivated at 22 °C was further subjected to plaque purification. The live-attenuated MERS-CoV vaccine strain resulting from this process was designated as MERS-CoV (EMC2012-CA 22 °C). The genomes of five purified clones were sequenced to identify any genetic changes compared to the original virus.

### 2.3. Sensitivity of Attenuated MERS-CoV Vaccine Virus to the Temperature

Vero cells grown in 6-well plates were inoculated with 0.001 multiplicity of infections (m.o.i) of live-attenuated MERS-CoV (EMC2012-CA 22 °C) and wild-type MERS-CoV (EMC2012). The inoculated cells were cultured in a incubator (5% CO_2_) at 37 °C or 41 °C for 4 days. Virus titers were measured by a log 10 tissue culture infectious dose of 50/mL (log_10_TCID_50_/mL) at 35 °C.

### 2.4. Quantification of Plaque-Forming Units (pfu)

A plaque assay was conducted to estimate the plaque-forming units (pfu) in the MERS-CoV stocks, EMC2012 or EMC2012-CA 22 °C. The viruses were diluted serially in MEM supplemented with 1.5% BSA and subsequently inoculated into Vero cells cultured in 6-well cell culture plates. The cells were infected for 4 h in a cell culture incubator (5% CO_2_, 35 °C), before the infected Vero cells were overlaid with an agar solution (1%) (LPS Solution, Republic of Korea) dissolved in MEM. After 4 days, the cells were stained with a 0.1% crystal violet solution (Sigma-Aldrich, St. Louis, MO, USA) dissolved in z formaldehyde solution (37%), and the plaque number was calculated.

### 2.5. Estimation of Viral Titers by log_10_TCID_50_/mL

The viral supernatants or viral tissue samples were diluted in a 10-fold serial manner using MEM supplemented with 1.5% BSA. These dilutions were then used to infect the fully grown Vero cells in 96-well plates. The infected cells were incubated in a incubator (5% CO_2_, 35 °C). After incubating the infected cells for 4 days, they were examined under a microscope for cytopathic effects (CPE). The viral titers, expressed as log10TCID50/mL, were determined based on the observed CPEs. The titers were determined as described by Muench and Reed [21].

### 2.6. Attenuation Confirmation of Live-Attenuated MERS-CoV Vaccine Virus in Animal

The K18-hDPP4 mice (*n* = 13 per group) were anaesthetized with isoflurane USP (Gujarat, India) and were then intranasally (i.n.) infected with 50 μL (2 × 10^4^ pfu) of the cold-adapted MERS-CoV vaccine virus, EMC2012-CA 22 °C, or wild-type MERS-CoV virus, EMC2012. Mice infected with PBS were used as controls. Following infection, the mice were closely observed for body weight changes and death. On day 6 post-infection (p.i.), the mice (*n* = 3 per group) were euthanized. Nasal turbinates, brain, lungs, and kidneys were collected as tissue samples for subsequent analysis. Each tissue sample weighing 0.1 g was homogenized in 1 mL of PBS (pH 7.4) using a BeadBlaster homogenizer (Benchmark Scientific, Edison, NJ, USA). The resulting homogenates were used to determine the viral titers, expressed as log_10_TCID_50_/g. The remaining tissue samples were reserved for histopathological examination.

### 2.7. Tissue Staining with Hematoxylin and Eosin

The fixation of each tissue was performed with neutral buffered formalin (10%) before it was embedded in paraffin. The sliced tissues (5 μm) were stained using a haematoxylin (H) solution for 90 s. Tap water was used on the H stained tissues and then the washed tissues were stained with an eosin (E) solution for 90 s. Pictures of the stained tissues were taken using an Olympus DP70 microscope (Olympus Corporation, Tokyo, Japan).

### 2.8. Vaccine Efficacy in K18-hDPP4 Mice

The K18-hDPP4 mice (*n* = 13) were i.n. immunized with the EMC2012-CA 22 °C vaccine virus (2 × 10^4^ pfu) and i.n. inoculated with Korean MERS-CoV (2 × 10^4^ pfu) 4 weeks after the vaccination. As controls, mice (*n* = 13) vaccinated with PBS and infected with MERS-CoV virus were used. Before the immunized mice were challenged with wild-type MERS-CoV (Korean MERS-CoV/2015), their sera were collected to measure the titers of the viral neutralizing antibody. The body weight changes and mortality of the challenged mice were recorded. Six days after challenge, three mice per group were euthanized to measure the virus titers in their nasal turbinate, brain, lung, kidney, and stained tissues (brain, lung, and kidney). Each tissue sample weighing 0.1 g was homogenized in 1 mL of PBS (pH 7.4) using a BeadBlaster homogenizer (Benchmark Scientific, Edison, NJ, USA). The viral titers were measured using log_10_TCID_50_/g.

### 2.9. Viral Neutralizing Antibody Titers

The sera (*n* = 10 per group) collected from the vaccinated K18-hDPP4 mice with 2 × 10^4^ pfu of EMC2012-CA 22 °C 4 weeks after the vaccination were 10-fold diluted in MEM supplemented with 1.5% BSA before they were serially two-fold diluted. The diluted sera (100 μL per sample) were incubated with 100 μL of 100TCID_50_/mL of wild-type MERS-CoV virus (Korean MERS-CoV/2015) for 1 h in a cell culture incubator (5% CO_2_, 37 °C). The warm PBS (pH 7.4) was used to wash the Vero cells before the mixed samples were inoculated. The inoculated Vero cells were cultured for 4 days before they were observed for CPEs. The titers of the viral neutralizing antibody were determined as 100% CPEs of the 4 wells in a given dilution of serum that was inhibited. The assay was carried out in quadruplicate.

### 2.10. Determining Lymphocytes Expressing Mouse IFN–γ

A total of six K18-hDPP4 mice were i.n. vaccinated with EMC2012-CA 22 °C (2 × 10^4^ pfu). Four weeks after the vaccination, the mice were euthanized prior to the collection of spleen samples. The collected spleen samples were homogenized in PBS (pH 7.4) and then overlaid with HISTOPAQUE-1077 (Sigma) prior to the centrifugation using a centrifuge (30 min, 1500 rpm, and 4 °C). The layer containing lymphocytes was harvested and used for the IFN-γ ELISpot assay with the Mouse IFN-γ ELISpotPlus kit (MABTECH, Nacka Strand, Sweden).

To prepare the assay plate, the wells were washed four times with 200 μL of PBS (pH 7.4) and stabilized with 200 μL of RPMI 1640 medium containing fetal bovine serum (FBS) (10%) for 30 min at 24 °C. The separated lymphocytes (250,000 cells/well) were infected with EMC2012-CA 22 °C at a multiplicity of infection (m.o.i) (0.01), and were allocated to each well of the plate. The plate was then incubated in a humidified incubator (5% CO_2_, 37 °C) for 24 h.

After incubation, the wells were washed five times with 200 μL of PBS (pH 7.4). A detection antibody (R4-6A2-biotin) was diluted in 1 μg/mL of PBS (pH 7.4) with 0.5% FBS (200 μL/well), and was then added before the plate was incubated for 2 h at 24 °C. Next, streptavidin-ALP (100 μL), diluted 1:1000 in PBS with 0.5% FBS, was added to each well of the plate after the plate was washed five times with PBS (pH 7.4) (200 μL). The plate was then incubated for 1 h at 24 °C and washed again five times with PBS (pH 7.4) (200 μL). The washed wells of the plate were added with 100 μL of the substrate solution (BCIP/NBT-plus) and developed until clear spots were detected. The spots were counted under a microscope (Olympus, Japan). Six K18-hDPP4 mice immunized with PBS mock were used as controls. Tissues including nasal turbinates, lungs, and kidneys were also collected to detect MERS-CoV-specific IgA antibodies.

### 2.11. Detection of Mouse Cytokines, TNF-α, IL-4, and IL-10, in Spleen Lymphocytes Using Enzyme-Linked Immunosorbent Assay

Splenocytes (2 × 10^6^/mL) used for the mouse IFN-γ ELISpot assay in RPMI 1640 medium with 10% FBS were stimulated with EMC2012-CA 22 °C (0.01 m.o.i ) in a humidified incubator (5% CO_2_, 37 °C) for 24 h, and the supernatants were collected to measure the TNF-α (Th1 cytokine), IL-4, and IL-10 (Th2 cytokines) using ELISA kits for mouse TNF-α, IL-4, and IL-10 (ThermoFisher Scientific, Waltham, MA, USA), following the manufacturer’s instructions. Briefly, the supernatant or standard (50 μL) was added to the wells of the plate, which were previously coated with cytokine antibodies prior to the incubation for 2 h at 24 °C. The wells were washed six times with the wash buffer (200 μL) and then streptavidin-HRP (100 μL) was added before the plate was incubated for 1 h at 24 °C. The wells were washed with the wash buffer and then TMB (100 μL) was added before the wells were incubated for 30 min at 24 °C. The stop solution (100 μL) was added to the reaction to stop the reaction, and the absorbance was detected at 450 nm using a spectrophotometer (Bio-Rad, Hercules, CA, USA). The concentration of the cytokines was calculated according to the standard curve.

### 2.12. Measurement of MERS-CoV-Specific IgA Antibody in Tissues of Vaccinated Mice

The wells of a Nunc-Immuno plate immune plate (Sigma-Aldrich) were coated with purified, inactivated MERS-CoV (EMC2012-CA 22 °C) antigen diluted to 100 μg/mL in a carbonate–bicarbonate buffer (pH 9.6) overnight at 4 °C. The plates were washed two times with 400 μL of PBS (pH 7.4), with horse serum (4%) and Tween 20 (0.05%) (Sigma). Each well of the plate was blocked with 400 μL of blocking buffer containing PBS and 4% skimmed milk over 12 h at 4 °C. After the buffer was discarded, each well of the plate had 100 μL (diluted 10-fold in blocking buffer) of the homogenized tissue supernatants of the tissues (nasal turbinates, lungs, and kidneys) from the vaccinated mice or PBS-mock vaccinated mice added, and was incubated for 1 h at 24 °C. Goat anti-mouse IgA secondary antibody HRP (100 μL, Invitrogen, Waltham, MA, USA), diluted to 1:5000 in blocking buffer, was also added to the wells, which were incubated for 1 h at 24 °C. The wells of the plate were washed 4 times with the wash buffer and then 100 μL of goat anti-mouse IgA secondary antibody HRP (Invitrogen, Waltham, MA, USA), diluted to 1:5000 in blocking buffer, was added before the plate was incubated for 1 h at 24 °C. The wells of the plate were washed 4 times with the wash buffer and then 100 μL of TMB ELISA substrate (MABTECH) was added before the plate was incubated for 30 min at 24 °C. In total, 100 μL of ABTS^®^ Peroxidase Stop Solution (KPL, Gaithersburg, MD, USA) was added to stop the reaction. The absorbance was detected at 450 nm with an iMARK^™^ Microplate Absorbance Reader (Bio-Rad, Hercules, CA, USA).

### 2.13. Genome Decoding of Live-Attenuated MERS-CoV Vaccine Virus

RNA genomes in five clones of MERS-CoV (EMC2012-CA 22 °C) were sequenced with overlapping primers (Appendix A). Viral RNA was purified with the RNeasy Mini Kit (QIAGEN, Venlo, The Netherlands). Viral stock (200 μL) grown in Vero cells with MEM medium was lysed with 350 μL of Buffer RLT before 500 μL of 70% ethanol was added to the disrupted mixture. The reacted sample (700 μL) was moved to the RNeasy Mini spin column and then the column was spun down in a microcentrifuge (15 s, 13,500 rpm). The flow-through was thrown out, and RW1 buffer (700 μL) was added to the spin column before the centrifugation (15 s, 13,500 rpm). The flow-through was removed and then RPE buffer (500 μL) was added to the spin column prior to the centrifugation (2 min, 13,500 rpm). The spin column was transferred to a new 1.5 mL tube prior to the elution of the viral RNA with RNAse-free water (50 μL). The eluted RNA was reverse-transcribed to cDNAs using a GoScript™ Reverse Transcription System (Promega, Madison, WI, USA) and 12 reverse primers (Appendix A). The cDNAs were PCR amplified with a segment-specific primer set (Appendix A) using GoTaq Hot Start Green Master Mix (Promega). The PCR products were run on the agarose gel and extracted with the QIAquick Gel Extraction Kit (QIAGEN). The purified genes were ligated into the pGEM-T Easy vector (Promega) by transformation into competent DH5α cells (Enzynomics, Daejeon, Republic of Korea). The successfully cloned plasmids were purified with the HiGene Plasmid Mini Prep Kit (BIOFACT, Daejeon, Republic of Korea) before the sequence was performed. Five clones were decoded. DNASTAR Lasergene (Madison, WI, USA) was used to analyze the genes. The sequence of MERS-CoV (EMC2012-CA 22 °C) was deposited into GenBank with the accession number of OQ533494.

### 2.14. Ethical Approval

The internal animal use committee at Chungnam National University (CNU) approved the animal study (202012A-CNU-174). The mouse works were performed according to the Korean legal guidelines and animal use regulations.

### 2.15. Statistical Analysis

The statistical analysis on the results among the animal groups were performed with Student’s *t*-test (IBM SPSS Statistics version 20). A *p* value (*p* < 0.05) was considered to be statistically significant.

## 3. Results

### 3.1. Development of Live-Attenuated MERS-CoV Vaccine Virus by Cold Adaptation

The pathogenicity of the attenuated MERS-CoV (EMC2012-CA 22 °C) strain was evaluated in the K18-hDPP4 mice. A total of 2 groups of K18-hDPP4 mice, with 13 mice per group, were intranasally infected with either 2 × 10^4^ pfu (50 μL in PBS) of the cold-adapted MERS-CoV (EMC2012-CA 22 °C) or the wild-type MERS-CoV (EMC2012) strain. The infected K18-hDPP4 mice were monitored for mortality (Figure 1A) and changes in their body weight over a 14-day period (Figure 1B). The results showed that the K18-hDPP4 mice infected with the cold-adapted MERS-CoV (EMC2012-CA 22 °C) did not experience any death and did not show significant weight loss. In contrast, the K18-hDPP4 mice infected with the wild-type MERS-CoV (EMC2012) strain exhibited 100% mortality within 8 days post-infection and experienced a weight loss of up to 1.2% (Figure 1A,B). The PBS-mock uninfected K18-hDPP4 mice did not show any death or weight loss.

To assess the viral replication, the viral titers in various tissues (nasal turbinates, brain, lungs, and kidneys) were measured on day 6 post-infection. The viral titers were quantified as log_10_TCID50/g in the Vero cells (Figure 1C). High viral titers were detected in the tissues of the K18-hDPP4 mice (*n =* 3) infected with the wild-type MERS-CoV strain, whereas the tissues of the K18-hDPP4 mice (*n =* 3) infected with the cold-adapted MERS-CoV strain did not show viral titers, which means they were below the detection limit of 1 log_10_TCID50/g. Specifically, the viral titers in the nasal turbinates, brain, lungs, and kidneys of the K18-hDPP4 mice infected with the wild-type MERS-CoV strain were 4.5, 6.5, 5.0, and 2.5 log_10_TCID50/g, respectively (Figure 1C).

To evaluate the tissue damage, brain, lung, and kidney tissues from the infected K18-hDPP4 mice were stained with H&E (Figure 2). In the brain tissue of the K18-hDPP4 mice infected with the wild-type MERS-CoV strain, lymphocytic perivascular cuffing was observed (Figure 2B). However, no lymphocytic perivascular cuffing was observed in the brain tissue of the K18-hDPP4 mice infected with the cold-adapted MERS-CoV strain (Figure 2C) or in the brain tissue of the PBS-mock uninfected K18-hDPP4 mice (Figure 2A).

Similarly, interstitial lymphocytes were detected in the kidney tissue of the K18-hDPP4 mice infected with the wild-type MERS-CoV strain (Figure 2E), while no lymphocyte infiltration was observed in the kidney tissue of the K18-hDPP4 mice infected with the cold-adapted MERS-CoV strain (Figure 2F) or in the kidney tissue of the PBS-mock uninfected K18-hDPP4 mice (Figure 2D).

In the lung tissue, interstitial pneumonia with lymphocyte infiltration was observed in the K18-hDPP4 mice infected with the wild-type MERS-CoV strain (Figure 2H). However, no pneumonia was observed in the lung tissue of the K18-hDPP4 mice infected with the cold-adapted MERS-CoV strain (Figure 2I) or in the lung tissue of the PBS-mock uninfected K18-hDPP4 mice (Figure 2G).

### 3.2. Temperature Sensitivity of Cold-Adapted MERS-CoV Vaccine Strain

The sensitivity of the cold-adapted MERS-CoV vaccine strain (EMC2012-CA 22 °C) was determined at 37 °C and 41 °C in the Vero cells by log_10_TCID_50_/mL (Figure 3A). At 37 °C, the viral titers of the cold-adapted MERS-CoV and wild-type MERS-CoV were 1.5 and 5.5 log_10_TCID50/mL, respectively. At 41 °C, the viral titer of the cold-adapted MERS-CoV was not defected, while the viral titer of the wild-type MERS-CoV was 5.0 log_10_TCID_50_/mL.

### 3.3. Induction of Mucosal Antibody (IgA) in the Tissues of K18-hDPP4 Mice Immunized with Cold-Adapted MERS-CoV Vaccine Strain

In the study, the induction of IgA antibodies was measured in the tissues (nasal turbinates, lungs, and kidneys) collected from the K18-hDPP4 mice 4 weeks after intranasal immunization with 2 × 10^4^ pfu of the cold-adapted MERS-CoV (EMC2012-CA 22 °C) vaccine strain (Figure 3B). The level of IgA antibodies was quantified by measuring the optical density (OD) using purified inactivated MERS-CoV antigen (EMC12) and goat horseradish peroxidase (HRP)-labeled anti-mouse IgA antibody.

The results showed that MERS-CoV-specific IgA antibodies were detected in the nasal turbinates, lungs, and kidneys of the immunized mice. Among these tissues, the lung tissue exhibited the highest level of IgA antibodies, with an OD value of 0.4. We evaluated the IgA titers in the kidneys since MERS-CoV could infect those of K18-hDPP4 mice. This indicates that the immunization with the cold-adapted MERS-CoV vaccine strain led to the production of specific IgA antibodies in the mucosal tissues, particularly in the lungs [22].

### 3.4. Induction of Cellular Immunity and Th1-Type Specific Cytokines in Spleen Lymphocytes of K18-hDPP4 Mice Immunized with Cold-Adapted Attenuated MERS-CoV Vaccine Strain

We measured the number of lymphocytes expressing IFN-γ in the spleens of the K18-hDPP4 mice immunized with the MERS-CoV vaccine strain (EMC2012-CA 22 °C) to find out whether this vaccine virus could induce cellular immunity, which is important for controlling virus infections in hosts (Figure 3C). The number of lymphocytes expressing IFN-γ in the splenocytes from the K18-hDPP4 mice, which were immunized with 2 × 10^4^ pfu of the EMC2012-CA 22 °C vaccine strain 4 weeks previously, was 1750/250,000, while that in the PBS-mock vaccinated K18-hDPP4 mice was 220/250,000.

### 3.5. Measurement of Th1 and Th2 Cytokines

We also measured the amount of Th1 cytokine (TNF-α) and Th2 cytokines (IL-4 and IL-10) in the supernatant of the lymphocytes from the spleens of the K18-hDPP4 mice immunized with the MERS-CoV vaccine strain (EMC2012-CA 22 °C) using ELISA (Figure 3D). The Th1 cytokine, TNF-α, was dominantly induced with the amount of 48 pg/mL, but the Th2 cytokines (IL-4 and IL-10) were not induced.

### 3.6. Protection of K18-hDPP4 Mice Immunized by Cold-Adapted MERS-CoV Vaccine Strain

The K18-hDPP4 mice (*n =* 13) were immunized with a single dose of the MERS-CoV vaccine strain (EMC2012-CA 22 °C) at a dosage of 2 × 10^4^ plaque-forming units (pfu). The aim was to evaluate the protective efficacy of the vaccine. Four weeks after vaccination, sera were collected from a subset of immunized K18-hDPP4 mice (*n =* 10), and their neutralizing antibody (NA) titers against the Korean MERS-CoV/2015 strain were measured using Vero cells (Figure 4A). Robust NA titers were observed in the immunized mice, ranging from 1280 to 5120.

On the fourth week after vaccination, the immunized K18-hDPP4 mice and sham-immunized controls (*n =* 13 per group) were intranasally challenged with 2 × 10^4^ pfu of the Korean MERS-CoV/2015 strain (Figure 4B,C). The challenged mice were monitored for mortality (Figure 4B) and changes in their body weight (Figure 4C) over a 14-day period. All the immunized and challenged K18-hDPP4 mice survived without any significant loss in body weight, while all the unimmunized and challenged K18-hDPP4 mice died within eight days of the challenge, experiencing a 2.2% loss in body weight (Figure 4B,C).

On the sixth day after the challenge, the viral titers were measured in various tissues (nasal turbinates, brain, lungs, and kidneys) collected from the challenged K18-hDPP4 mice (Figure 4D). No virus was detected in the tissues of the immunized and challenged mice. However, high viral titers, ranging from 2.0 (kidney) to 6.0 (brain) log_10_TCID50/g, were observed in the tissues of the unimmunized and challenged mice.

The collected tissues (brain, lung, and kidney) from the challenged K18-hDPP4 mice were subjected to hematoxylin and eosin (H&E) staining (Figure 5). Lymphocytic perivascular cuffing was observed in the brain veins of the wild-type MERS-CoV-infected K18-hDPP4 mice (Figure 5B). In contrast, no lymphocytic perivascular cuffing was observed in the brain veins of the K18-hDPP4 mice infected with the cold-adapted MERS-CoV vaccine strain (Figure 5C), or in the brain veins of the PBS-mock uninfected K18-hDPP4 mice (Figure 5A). Similarly, interstitial lymphocyte infiltration was observed in the kidneys of the K18-hDPP4 mice infected with wild-type MERS-CoV (Figure 5E), whereas no lymphocyte infiltration was detected in the kidneys of the K18-hDPP4 mice infected with the cold-adapted MERS-CoV vaccine strain (Figure 5F), or in the kidneys of the PBS-mock uninfected K18-hDPP4 mice (Figure 5D). Furthermore, interstitial pneumonia and lymphocyte infiltration were detected in the lungs of the K18-hDPP4 mice infected with wild-type MERS-CoV (Figure 5H), while no pneumonia was observed in the lungs of the K18-hDPP4 mice infected with the cold-adapted MERS-CoV vaccine strain (Figure 5I), or in the lungs of the PBS-mock uninfected K18-hDPP4 mice (Figure 5G).

### 3.7. Decoding the Genome Sequence of the Cold-Adapted Live-Attenuated MERS-CoV Vaccine Strain

We performed gene sequencing of the cold-adapted live-attenuated MERS-CoV vaccine strain (EMC2012-CA 22 °C) and compared it with the wild-type MERS-CoV strain (EMC2012) (GenBank accession number: NC_019843.3). Based on sequenced five clones of vaccine strains, we found the common changes in the five clones. Among the total of the 12,987 amino acids (AA) examined, 16 AA changes were observed in the genes encoding the open reading frame (ORF)1a protein, S protein, and M protein of the MERS-CoV vaccine strain (EMC2012-CA 22 °C) (Table 1, Appendix A). Specifically, nine AA changes occurred in the ORF1a polyprotein, six AA changes in the S protein, and one AA change in the M protein of the MERS-CoV vaccine strain (EMC2012-CA 22 °C). Notably, none of the AA changes affected the receptor-binding domain (RBD) within the MERS-CoV spike protein (AA 367-606) [23]. Additionally, one AA change (H79R) was observed in the non-structural protein 1 (nsp1), which is involved in suppressing the host immune response and reducing the host gene expression [24]. Furthermore, three AA changes (H1616L, T2088P, and A2210V) were identified in nsp3, which plays a crucial role in the coronavirus replication and transcription complex [25]. Additionally, one AA change (Q3295R) was found in nsp5, which is involved in cleaving coronaviral polypeptide 1a/1ab to yield the mature nsp4-6 of coronavirus [26]. Two AA changes (F3735I and E3822G) were observed in nsp6, which antagonizes the pI interferon response in infected cells [27]. One AA change (P4195S) was detected in nsp9, an essential accessory component of the coronavirus replication and transcription complex [28]. Lastly, one AA change (N4247S) occurred in nsp10, which acts as a viral exonuclease to decrease the mutation rate of the coronavirus’s error-prone RNA-dependent RNA polymerase [29]. Additionally, one AA change (T5M) was found in the M protein, a structural component of MERS-CoV.

## 4. Discussion

MERS-CoV is a zoonotic virus that can cause severe diseases in humans. Although this virus primarily circulates among camels in countries on the Arabian Peninsula, its potential to be transmitted to humans highlights the need for an effective vaccine to protect against MERS-CoV infections. Our research focused on developing a live-attenuated MERS-CoV vaccine through cold adaptation at 22 °C in Vero cells, which could be administered intranasally to humans. The vaccine strain we developed, known as EMC2012-CA 22 °C, exhibited a high attenuation in K18-hDPP4 mice, and a single dose provided complete protection against MERS-CoV infections in these mice.

Our findings revealed that the MERS-CoV vaccine strain (EMC2012-CA 22 °C) was sensitive to temperatures of 37 °C and 41 °C. Currently, the licensed human vaccine utilizing cold adaptation is the seasonal human influenza vaccine, which is delivered through nasal spray. This nasal spray influenza vaccine was developed by adapting influenza viruses (A&B) to replicate in primary chicken kidney cells and fertilized eggs up to 25 °C [30,31].

When we administered the MERS-CoV vaccine strain (EMC2012-CA 22 °C) intranasally to the K18-hDPP4 mice, it induced a strong neutralizing antibody response against the virus. Furthermore, it protected the mice from lethal challenges by wild-type MERS-CoV, as evidenced by the absence of mortality, body weight loss, and viral detection in various tissues such as nasal turbinates, brains, lungs, and kidneys. Multiple candidate MERS-CoV vaccines have been evaluated for their efficacy in K18-hDPP4 mice [32,33,34,35,36]. In one study, robust neutralizing antibodies were induced in mice and non-human primates immunized with a DNA vector encoding the S gene of MERS-CoV, along with the expressed S1 protein of MERS-CoV [32]. Moreover, immunized non-human primates were protected against MERS-CoV challenges that typically lead to pneumonia. Another study explored the potential of a measles virus (MV) expressing the S genes of MERS-CoV as a vaccine candidate [33]. Immunized mice lacking the type I interferon receptor (CD46Ge mice) exhibited a strong induction of both MV and MERS-CoV neutralizing antibodies, and they were protected from MERS-CoV challenges without developing pneumonia in their lungs. Rhesus macaques immunized with a synthetic DNA vaccine encoding the S gene of MERS-CoV displayed reduced clinical symptoms, viral loads, and pathological signs in their lungs compared to control animals [34]. Mice that received intranasal immunization with recombinant parainfluenza virus 5 expressing the S gene of MERS-CoV were also protected from fatal MERS-CoV infections [35]. Furthermore, live-attenuated vaccines developed through genetic modification of the E gene of MERS-CoV conferred complete protection against lethal MERS-CoV challenges in immunized mice [36].

The immunization of the K18-hDPP4 mice with the MERS-CoV vaccine strain (EMC2012-CA 22 °C) induced MERS-CoV-specific IgA antibodies. IgA antibodies play a crucial role in defending the respiratory and gastrointestinal tracts against pathogen invasions in humans and animals [37].

Concerning the reversion of our cold-adapted MERS-CoV vaccine strain into the wild-type virus, further studies are clearly warranted to determine the likelihood of this reversion to virulence after passage in cell culture or animals. In addition, an efficacy study on camels, long-term B and T cell immune responses in animals, and human clinical tests are needed.

## 5. Conclusions

MERS-CoV, a zoonotic virus causing severe diseases, necessitates an effective vaccine. Our research developed a live-attenuated MERS-CoV vaccine, EMC2012-CA 22 °C, which provided complete protection in mice and induced neutralizing antibodies. The MERS-CoV vaccine induced MERS-CoV-specific IgA antibodies, which are crucial for respiratory and gastrointestinal defense.

## Figures and Tables

**Figure 1 vaccines-11-01353-f001:**
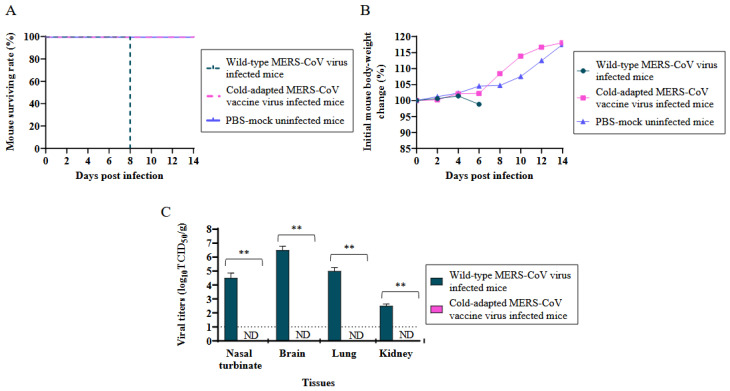
Attenuation confirmation of cold adapted MERS-CoV vaccine strain in mice. K18-hDPP4 mice (*n =* 13 per group) were i.n. infected with 2 × 10^4^ pfu of MERS-CoV vaccine strain (EMC2012-CA 22 °C) or were i.n. infected with 2 × 10^4^ pfu of wild-type MERS-CoV strain (EMC2012). The infected mice were observed for mortality and body weight change for 14 days. PBS-mock infected hDPP4-mice (*n =* 13) were used for the control group. Mice (*n =* 3 per group) were euthanized on day 6 p.i. for the viral titration in tissues (nasal turbinate, brain, lung, and kidney). (**A**) Mouse mortality (%); (**B**) the change in body weights (%) of mice compared to the body weights before infections; and (**C**) viral titers in tissues (log_10_TCID50/g, *n =* 3 per group). The viral titers are the mean of 3 tissues ± standard deviation. Dotted line (....): limit of detection. ** *p* < 0.001, ND: non-detected.

**Figure 2 vaccines-11-01353-f002:**
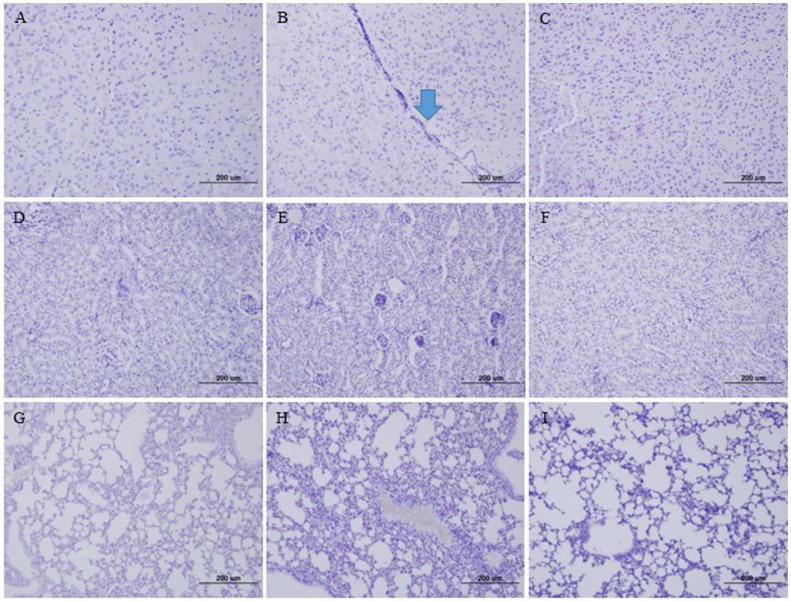
Tissue pathology in mice infected with cold-adapted MERS-CoV vaccine strain. Tissues collected for viral titrations in Figure 1 were stained with hematoxylin and eosin (×100). (**A**) Brain tissue of PBS-mock K18-hDPP4 mouse; (**B**) Brain tissue of wild-type MERS-CoV (EMC2012)-infected K18-hDPP4 mouse, arrow (lymphocyte infiltration); (**C**) Brain tissue of K18-hDPP4 mouse infected with cold-adapted MRES-CoV vaccine strain (EMC2012-CA 22 °C); (**D**) Kidney tissue of PBS-mock K18-hDPP4 mouse; (**E**) Kidney tissue of wild-type MERS-CoV (EMC2012)-infected K18-hDPP4 mouse; (**F**) Kidney tissue of K18-hDPP4 mouse infected with cold-adapted MRES-CoV vaccine strain (EMC2012-CA 22 °C); (**G**) Lung tissue of PBS-mock K18-hDPP4 mouse; (**H**) Lung tissue of wild-type MERS-CoV (EMC2012)-infected K18-hDPP4 mouse; and (**I**) Lung tissue of K18-hDPP4 mouse infected with cold-adapted MRES-CoV vaccine strain (EMC2012-CA 22 °C).

**Figure 3 vaccines-11-01353-f003:**
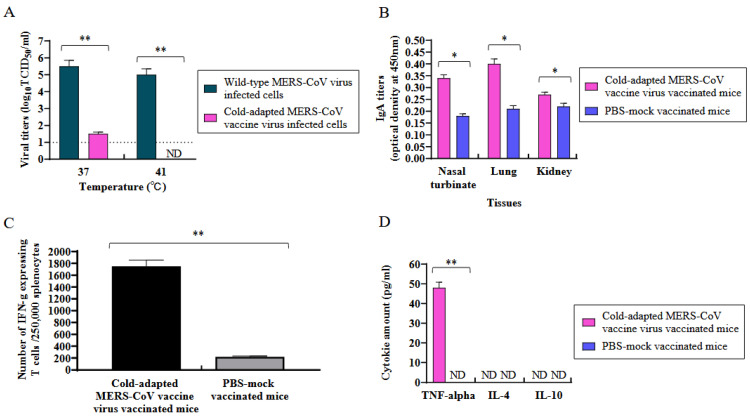
Temperature sensitivity, mucosal IgA antibody induction, cellular immunity, and cytokine profiles of cold-adapted attenuated MERS-CoV vaccine strain in mice. (**A**) Temperature sensitivity of cold-adapted attenuated MERS-CoV vaccine strain. Vero cells were infected with 0.001 m.o.i of MERS-CoV (EMC2012-CA 22 °C) or wild-type MERS-CoV (EMC2012) and were incubated at 37 °C or 41 °C for 4 days. The viral titers in the supernatants were measured by log10TCID50/mL. The viral titers are the mean of 3 samples ± standard deviation. Dotted line (....): limit of detection. ** *p* < 0.001, ND: non-detected. (**B**) MERS-CoV-specific IgA antibody induction in tissues of immunized mice. K18-hDPP4 mice (*n =* 3) infected with 2 × 104 pfu of MERS-CoV vaccine strain (EMC2012-CA 22 °C) were euthanized at 4 weeks p.i. and tissues (nasal turbinate, lung, and kidney) were collected for measurement of IgA antibody using ELISA with the plate coated with purified inactivated MERS-CoV (EMC2012-CA 22 °C) antigen. The IgA titers are the mean of 3 tissues ± standard deviation. * *p* < 0.05. (**C**) Number of lymphocytes expressing IFN-γ in spleen cells of the immunized mice. K18-hDPP4 mice (*n =* 6) vaccinated with 2 × 104 pfu of EMC2012-CA 22 °C were euthanized at 4 weeks after immunization for spleen collection. The purified lymphocytes were used to measure the number of lymphocytes expressing IFN-γ using ELISPT assay. The number of lymphocytes are the mean of 3 reactions ± standard deviation. ** *p* < 0.001. (**D**) Measurement of cytokines (Th1, Th2) in splenocytes from immunized mice. Splenocytes (2 × 106/mL) used for mouse IFN-γ ELISpot assay were stimulated with EMC2012-CA 22 °C (0.01 m.o.i ) for 24 h. The supernatants were collected to measure Th1 cytokine (TNF-α) and Th2 cytokines (IL-4 and IL-10) using ELISA kits. The cytokine amount is the mean of 3 reactions ± standard deviation. ** *p* < 0.001.

**Figure 4 vaccines-11-01353-f004:**
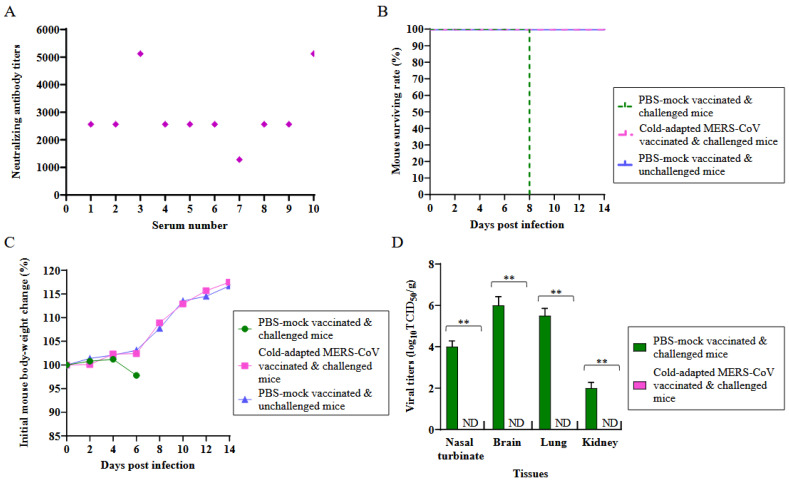
Induction of neutralizing antibody and challenge of the immunized mice with cold-adapted attenuated MERS-CoV vaccine strain. K18-hDPP4 mice (*n =* 13 per group), which were i.n. immunized with 2 × 10^4^ pfu of MERS-CoV vaccine strain (EMC2012-CA 22 °C) were i.n. challenged with 2 × 10^4^ pfu of wild-type MERS-CoV strain (Korean MERS-CoV/2015) at 4 weeks after vaccination. Sera were collected before the challenge and their neutralizing antibody was measured against wild-type MERS-CoV strain (Korean MERS-CoV/2015). The challenged K18-hDPP4 mice were monitored for mortality and body weight change for 14 days. PBS-mock challenged K18-hDPP4 mice (*n =* 13) were used for the control group. The infected K18-hDPP4 mice (*n =* 3 per group) were euthanized on day 6 after challenge to measure the viral titration in tissues (nasal turbinate, brain, lung, and kidney). (**A**) Neutralizing antibody titers in sera; (**B**) Mouse mortality of challenged K18-hDPP4 mice (%); (**C**) The change in body weights (%) of challenged K18-hDPP4 mice compared to the body weights before challenge; and (**D**) Viral titers in terms of log_10_TCID50/g in tissues (*n =* 3 per group) in nasal turbinates (0.1 g), brains (0.1 g), lungs (0.1 g), and kidneys (0.1 g) of challenged K18-hDPP4 mice. The viral titers are the mean of 3 tissues ± standard deviation. Dotted line (....): limit of detection. ** *p* < 0.001.

**Figure 5 vaccines-11-01353-f005:**
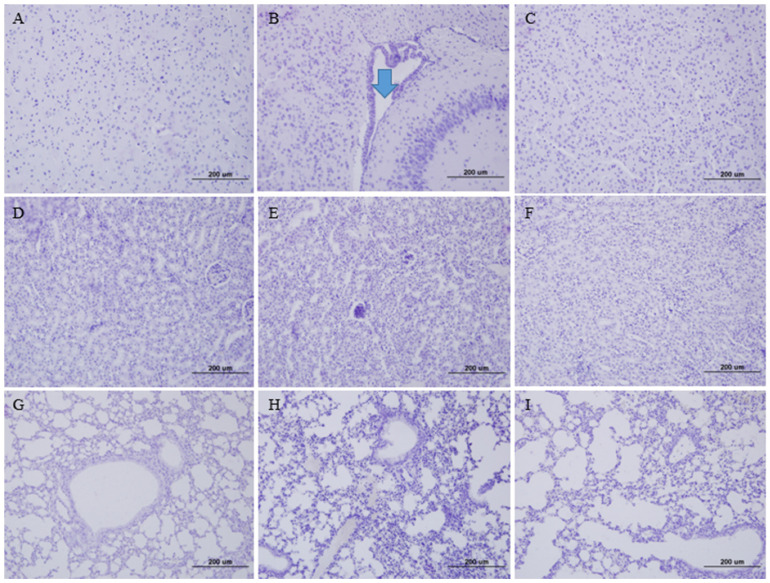
Tissue pathology in the challenged mice. Tissues collected for viral titrations in Figure 4D were stained with hematoxylin and eosin (×100). (**A**) Brain tissue of PBS-mock K18-hDPP4 mouse; (**B**) Brain tissue of the unimmunized and challenged K18-hDPP4 mouse, arrow (lymphocyte infiltration); (**C**) Brain tissue of the immunized and challenged K18-hDPP4 mouse; (**D**) Kidney tissue of PBS-mock hDPP4 mouse; (**E**) Kidney tissue of the unimmunized and challenged K18-hDPP4 mouse; (**F**) Kidney tissue of the immunized and challenged K18-hDPP4 mouse; (**G**) Lung tissue of PBS-mock hDPP4 mouse; (**H**) Lung tissue of the unimmunized and challenged K18-hDPP4 mouse; and (**I**) Lung tissue of the immunized and challenged K18-hDPP4 mouse.

**Table 1 vaccines-11-01353-t001:** The changed amino acids in cold-adapted live-attenuated MERS-CoV vaccine strain (EMC2012-CA 22 °C).

Protein Name		Changed Amino Acid Sequences	The Number of Changed Sequence
ORF1a polyprotein	nsp1	H79R	1/193
	nsp3	H1616L, T2088P, and A2210V	3/1887
	nsp5	Q3295R	1/306
	nsp6	F3735I and E3822G	2/292
	nsp9	P4195S	1/110
	nsp10	N4247S	1/140
S protein		T38P, N66Y, S305R, T872A, I879T, and S1251F	6/1354
M protein		T5M	1/220
Total amino acids		16/12, 987 (including non-changed ORFs)	

## Data Availability

Data are available upon being requested.

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
