# Peer review of "The Protective Efficacy of Single-Dose Nasal Immunization with Cold-Adapted Live-Attenuated MERS-CoV Vaccine against Lethal MERS-CoV Infections in Mice"

_vaccines, 2023, doi:10.3390/vaccines11081353_

Round 1

Reviewer 1 Report

This manuscript describes generation of cold-adapted MERS-CoV and subsequent testing as a vaccine using DPP4-transgenic mice.  This is an interesting approach, although I am highly skeptical that such a vaccine will ever be approved because for the possibility of reversion to virulence, something that very much should be addressed directly in this manuscript.  For example, FluMist influenza vaccine can revert to a virulent phenotype.  The issue is that seasonal influenza does not have a 35% mortality rate.

Overall, this is an interesting strategy for vaccine development and the approaches used to characterize this candidate vaccine are appropriate and the studies appear to have been competently executed.  Clearly, it is impressive that a single dose of the vaccine elicited protective immunity that protected susceptible mice.  Nonetheless there are some fundamental issues that need to be addressed.

I was not able to determine with confidence whether the virus titers for EMC2012-CA22C were always done at reduced temperature or at 37.  Please carefully clarify this.

Most prominently, it appears that 5 clones of EMC2012-CA22C were sequenced but the description presented under section 3.7 is quite vague: you report a total of 16 AA changes, which is presumably among the 5 clones, but fail to mention whether any or some of these changes were common to the different cold-adapted clones.  This is important because it could provide clues to the basis of cold-attenuation and is also relevant to the problem of reversion to virulence.  I am assuming that you did not sequence any of the viruses after replication at 37C, which again is relevant to reversion.

In the discussion section, you make several totally unjustified or unwarranted statements:

“The MERS-CoV live attenuated vaccine (EMC2012-CA22°C) can be administered to healthy individuals under the age of 50 who are not pregnant.” – this sounds like you candidate is already approved.  Similarly, you soon write “The live attenuated vaccine can be given to healthy individuals under 50, and the strain has potential for producing inactivated split vaccines, benefiting pregnant women, immunocompromised individuals, and the elderly.”, which again is simply misleading. Suggest deleting both of these statements.

Line 16: no traces of [infectious] virus

Line 131: I think you mean MERS-CoV not SARS-CoV

Line 135+: Please rephrase this sentence – it appears that you tested in quadruplicate, but the “100% of cells in 96 well plate” is probably not correct (100% of the 4 wells representing a given dilution of serum).

Line 139: it appears as 2 x 104 PFU (need superscript to get 2 x 10^4)

Line 203: Odd font to reference Qiagen location

Line 204: suggest lysed rather than ruptured

Line 261: suggest changing “Viral titers by log10 TCID50/g in tissues (n=3 per group) of nasal turbinate” to “Viral titers in tissues (log10 TCID50/g, n=3 per group)”.

Suggest explaining the focus on kidneys relative to MERS-CoV (why would kidneys be assayed for IgA – I personally do not know).

Line 354: “On the fourth week after vaccination, the immunized K18-hDPP4 mice AND sham-immunized controls (n=13 per group)

Line 482: “Intranasal administration in K18-hDPP4 mice yielded promising results.” Is a meaningless statement – suggest deleting.

There are numerous small mistakes in the manuscript that need to be corrected:

-       Reference 18 is actually the last few characters of reference 17, which totally mixes up reference citations and needs to be fixed, and all citations should be confirmed as correct.

-       In numerous instances, you fail to indicate that titers are log10 (e.g. lines 249, 251, 294, 296 and elsewhere)

-       Line 380: p < 0.00. (??)

Not bad - a few suggestions offered.

Author Response

[Responses to reviewer’s comments]

Reviewer 1

Comments and Suggestions for Authors

This manuscript describes generation of cold-adapted MERS-CoV and subsequent testing as a vaccine using DPP4-transgenic mice.  This is an interesting approach, although I am highly skeptical that such a vaccine will ever be approved because for the possibility of reversion to virulence, something that very much should be addressed directly in this manuscript.  For example, FluMist influenza vaccine can revert to a virulent phenotype.  The issue is that seasonal influenza does not have a 35% mortality rate.

Overall, this is an interesting strategy for vaccine development and the approaches used to characterize this candidate vaccine are appropriate and the studies appear to have been competently executed.  Clearly, it is impressive that a single dose of the vaccine elicited protective immunity that protected susceptible mice.  Nonetheless there are some fundamental issues that need to be addressed.

I was not able to determine with confidence whether the virus titers for EMC2012-CA22C were always done at reduced temperature or at 37.  Please carefully clarify this.

Most prominently, it appears that 5 clones of EMC2012-CA22C were sequenced but the description presented under section 3.7 is quite vague: you report a total of 16 AA changes, which is presumably among the 5 clones, but fail to mention whether any or some of these changes were common to the different cold-adapted clones. 

[Response] We clarified it in the revised manuscript: lines 390-391:  Based on sequenced five clones of vaccine strains, we found out the common changes in the five clones.

This is important because it could provide clues to the basis of cold-attenuation and is also relevant to the problem of reversion to virulence.  I am assuming that you did not sequence any of the viruses after replication at 37C, which again is relevant to reversion.

[Response] We discussed it in the revised manuscript: lines 454-456: Concerning the reversion of our cold-adapted MERS-CoV vaccine strain into wild-type virus, the further study is warranted to confirm whether the reversion may occur in the cell culture at 37oC or vaccinated animals.

In the discussion section, you make several totally unjustified or unwarranted statements:

“The MERS-CoV live attenuated vaccine (EMC2012-CA22°C) can be administered to healthy individuals under the age of 50 who are not pregnant.” – this sounds like you candidate is already approved.  Similarly, you soon write “The live attenuated vaccine can be given to healthy individuals under 50, and the strain has potential for producing inactivated split vaccines, benefiting pregnant women, immunocompromised individuals, and the elderly.”, which again is simply misleading. Suggest deleting both of these statements.

[Response] We deleted it in the revised manuscript.

Line 16: no traces of [infectious] virus

[Response] We changed it in the revised manuscript: line 16: no traces of infectious virus

Line 131: I think you mean MERS-CoV not SARS-CoV

[Response] We changed it in the revised manuscript: line 131: wild-type MERS-CoV

Line 135+: Please rephrase this sentence – it appears that you tested in quadruplicate, but the “100% of cells in 96 well plate” is probably not correct (100% of the 4 wells representing a given dilution of serum).

[Response] We changed it in the revised manuscript: lines 135-137: determined as 100% CPE of the 4 wells representing a given dilution of serum was inhibited

Line 139: it appears as 2 x 104 PFU (need superscript to get 2 x 10^4)

[Response] We changed it in the revised manuscript: line 139: a dose of 2 × 104 pfu

Line 203: Odd font to reference Qiagen location

[Response] We changed it in the revised manuscript: line 203: (QIAGEN, Venlo, Netherlands).

Line 204: suggest lysed rather than ruptured

[Response] We changed it in the revised manuscript: line 204: was lysed

Line 261: suggest changing “Viral titers by log10 TCID50/g in tissues (n=3 per group) of nasal turbinate” to “Viral titers in tissues (log10 TCID50/g, n=3 per group)”.

[Response] We changed it in the revised manuscript: line 261: Viral titers in tissues (log10 TCID50/g, n=3 per group).

Suggest explaining the focus on kidneys relative to MERS-CoV (why would kidneys be assayed for IgA – I personally do not know).

[Response] We mentioned it in the revised manuscript: lines 296-297: We evaluated IgA titers in kidneys since MERS-CoV could infect them of K18-hDPP4 mice.

Line 354: “On the fourth week after vaccination, the immunized K18-hDPP4 mice AND sham-immunized controls (n=13 per group)

[Response] We changed it in the revised manuscript: lines 344-345: the immunized K18-hDPP4 mice and sham-immunized controls (n=13 per group)

Line 482: “Intranasal administration in K18-hDPP4 mice yielded promising results.” Is a meaningless statement – suggest deleting.

[Response] We deleted it in the revised manuscript.

There are numerous small mistakes in the manuscript that need to be corrected:

-       Reference 18 is actually the last few characters of reference 17, which totally mixes up reference citations and needs to be fixed, and all citations should be confirmed as correct.

[Response] We corrected the references  in the revised manuscript.

-       In numerous instances, you fail to indicate that titers are log10 (e.g. lines 249, 251, 294, 296 and elsewhere)

[Response] We changed it the revised manuscript: lines 249, 251, 284,285, 354: log10

-       Line 380: p < 0.00. (??)

[Response] We changed it the revised manuscript: line 370: **P <0.001

Reviewer 2 Report

Dr. Jang’s team developed a cold-adapted attenuated MERS-CoV vaccine candidate.  The nasally vaccinated mice showed the protection effect with lethal wide-type MERS-CoV viruses challenge.

It is a well-written manuscript but need some clarification:

Line 21 and 22, I think it is better to tone-down saying “believe”, there is no data shown by this study suggesting safety to human.  This study suggests the potential toward that goal but no experimental evidence showing that yet.

Line 49, please clarify the age of the animal when start the immunization, are they at similar range of age?

In Figure 2 and Figure 5, please label the tissue in each picture and make the arrow in B more notable (bigger or in red).

In the Discussion, it is better to address the potential reversion of the cold-adapted vaccine strain back to wide-type virulent virus.  Which AA changes may contribute to the cold adaption.

It would be better to address what will be the next step of the research:  future animal studies (in camel?), long term B and T cell immune response evaluation, further natural or artificial modification toward a safer vaccine in human.

In the Conclusions, the sentence of “The live attenuated vaccine can be given to…” is too strong and misleading.  It has the potential, but more studies are needed.  If the split vaccine approach is proposed to be used, why using this cold-adapted strain is beneficial?  The cold-adapted virus has mutations that are not found commonly in the wide, and is it better to use wide-type for split-virus vaccine?  Please clarify.

Author Response

[Responses to reviewer’s comments]

Reviewer 2

Dr. Jang’s team developed a cold-adapted attenuated MERS-CoV vaccine candidate.  The nasally vaccinated mice showed the protection effect with lethal wide-type MERS-CoV viruses challenge.

It is a well-written manuscript but need some clarification:

Line 21 and 22, I think it is better to tone-down saying “believe”, there is no data shown by this study suggesting safety to human.  This study suggests the potential toward that goal but no experimental evidence showing that yet.

[Response] We toned it down the revised manuscript: lines 21-22: the developed cold-adapted attenuated MERS-CoV vaccine strain can be one of candidates for human and animal vaccines.

Line 49, please clarify the age of the animal when start the immunization, are they at similar range of age?

[Response] We changed it the revised manuscript: lines 50-51: mice (K18-hDPP4) (5-week-old)

In Figure 2 and Figure 5, please label the tissue in each picture and make the arrow in B more notable (bigger or in red).

[Response] We changed it the revised manuscript in Figures 2 & 5.

In the Discussion, it is better to address the potential reversion of the cold-adapted vaccine strain back to wide-type virulent virus.  Which AA changes may contribute to the cold adaption.

[Response] We discussed it in the revised manuscript: lines 454-456: Concerning the reversion of our cold-adapted MERS-CoV vaccine strain into wild-type virus, the further study is warranted to confirm whether the reversion may occur in the cell culture at 37oC or vaccinated animals.

It would be better to address what will be the next step of the research:  future animal studies (in camel?), long term B and T cell immune response evaluation, further natural or artificial modification toward a safer vaccine in human.

[Response] We added it in the revised manuscript: lines 456-458: . In addition, the efficacy study in camels, long-term B and T cell immune responses in animals, and human clinical tests are needed. 

In the Conclusions, the sentence of “The live attenuated vaccine can be given to…” is too strong and misleading.  It has the potential, but more studies are needed.  If the split vaccine approach is proposed to be used, why using this cold-adapted strain is beneficial?  The cold-adapted virus has mutations that are not found commonly in the wide, and is it better to use wide-type for split-virus vaccine?  Please clarify.

[Response] We deleted this parts since reviewer 1 asked them to be deleted out.

Round 2

Reviewer 1 Report

The authors have addressed a large number of the issues I raised and I believe the manuscript has been significant improved.  A few small remaining issues warrant attention:

One non-trivial issue: is the limit of detection for virus in tissue homogenates really 1 log10 TCID50/GRAM (10 TCID50/GRAM) as indicated in Fig 4D??  I do not see a description of how much tissue was actually homogenized for assay, but do not think this can be correct.  A mouse brain weighs roughly 0.4 grams – if you did not assay 100 mg of each brain then your LOD would be > 10 TCID50/gram.  Please recheck these tissue LODs.

Lines 255-256, 369-70 (Figure 1 and 4 legend) – still have 2 x 104 instead of 2 x 10^4

There remains a lot of variability in describing virus titers in terms of log10TCID50, log10 TCID50, etc.  I think you should subscript 10 in log10.

Finally, thank you for addressing the important topic of reversion to virulence.  The text you have is “the further study is warranted to confirm whether the reversion may occur”.  This is acceptable to me, but I might suggest that a better way to phrase this would be “further studies are clearly warranted to determine the likelihood of reversion to virulence after passage in cell culture of animals.

Author Response

Responses to reviewer 1 (2nd )

The authors have addressed a large number of the issues I raised and I believe the manuscript has been significant improved.  A few small remaining issues warrant attention:

One non-trivial issue: is the limit of detection for virus in tissue homogenates really 1 log10 TCID50/GRAM (10 TCID50/GRAM) as indicated in Fig 4D??  I do not see a description of how much tissue was actually homogenized for assay, but do not think this can be correct.  A mouse brain weighs roughly 0.4 grams – if you did not assay 100 mg of each brain then your LOD would be > 10 TCID50/gram.  Please recheck these tissue LODs.

[Response] We clarified in the 2nd revised manuscript: lines 124-125 & 376: Each tissue sample weighing 0.1g was homogenized in 1 mL of PBS (pH 7.4) using a BeadBlaster homogenizer (Benchmark Scientific, Edison, New Jersey, USA).

nasal turbinate (0.1g), brain (0.1g), lungs (0.1g), and kidney (0.1g)

Lines 255-256, 369-70 (Figure 1 and 4 legend) – still have 2 x 104 instead of 2 x 10^4

[Response] We changed it in the 2nd revised manuscript: lines 252-253 & 366-367: 2 × 104

There remains a lot of variability in describing virus titers in terms of log10TCID50, log10 TCID50, etc.  I think you should subscript 10 in log10.

[Response] We changed it in the 2nd revised manuscript: lines 243, 246,248, 258, 292, 375, 361: log10TCID50/g

Finally, thank you for addressing the important topic of reversion to virulence.  The text you have is “the further study is warranted to confirm whether the reversion may occur”.  This is acceptable to me, but I might suggest that a better way to phrase this would be “further studies are clearly warranted to determine the likelihood of reversion to virulence after passage in cell culture of animals.

[Response] We changed it in the 2nd revised manuscript: lines 470-471: the further studies are clearly warranted to determine the likelihood of reversion to virulence after passage in cell culture or animals.
